

# Context-based sentiment analysis on customer reviews using machine learning linear models

Anandan Chinnalagu and  Ashok Kumar Durairaj

Computer Science, Government Arts College (Affiliated to Bharathidasan University, Tiruchirappalli), Kulithalai, Karur, Tamil Nadu, India

## ABSTRACT

Customer satisfaction and their positive sentiments are some of the various goals for successful companies. However, analyzing customer reviews to predict accurate sentiments have been proven to be challenging and time-consuming due to high volumes of collected data from various sources. Several researchers approach this with algorithms, methods, and models. These include machine learning and deep learning (DL) methods, unigram and skip-gram based algorithms, as well as the Artificial Neural Network (ANN) and bag-of-word (BOW) regression model. Studies and research have revealed incoherence in polarity, model overfitting and performance issues, as well as high cost in data processing. This experiment was conducted to solve these revealing issues, by building a high performance yet cost-effective model for predicting accurate sentiments from large datasets containing customer reviews. This model uses the fastText library from Facebook's AI research (FAIR) Lab, as well as the traditional Linear Support Vector Machine (LSVM) to classify text and word embedding. Comparisons of this model were also done with the author's a custom multi-layer Sentiment Analysis (SA) Bi-directional Long Short-Term Memory (SA-BLSTM) model. The proposed fastText model, based on results, obtains a higher accuracy of 90.71% as well as 20% in performance compared to LSVM and SA-BLSTM models.

## INTRODUCTION

Today, customer satisfaction plays a major role for a successful business providing products and/or services. An analysis of consumer reviews is crucial to understand what a customer wants in terms of sentiment, (*Duyu, Bing & Ting, 2015*) as well as the betterment of a company or business to grow overtime. The phrase "what other people think" has importance to a buyer's decision when purchasing products and services according to survey (*Pang & Li, 2008*). Most companies handle customer service *via* call-centers with live agents, and over the last few years, the availability of viewing opinions, reviews, testimonies, *etc.* (*Somasundaran et al., 2007*) on the web has provided new avenues of research for automatic subjectivity, understanding texts, and sentiment classification. Researchers would use ML and DL models with the use of Natural Language Processing (NLP) techniques to process

Corresponding author
Anandan Chinnalagu,
anandanc@hotmail.com

and classify datasets filled with various reviews in their study. However, the purpose of NLP is to analyze, extract, and present information for better decision-making in businesses. The level of granularity in the process of analyzing controversial texts vary from individual characters to sub-word units or words forming (*Conneau et al., 2017*) a sentence to sentences forming paragraphs. Early research and applied methods in text analysis discriminates at a sentence and phrase level (*Yu & Hatzivassiloglou, 2003*) between objective and subjective texts. Prime candidates for traditional solutions for a sentence level classification of a document include the bag-of-words approach, SVM's, or the Adaboost classifier. Accurate scores in sentiment, detection of negation and sarcasm, as well as challenges in word ambiguity and multi-polarization were taken into account when making considerations to machine learning algorithms while building a sentiment classifier. The evaluation of the models includes audio transcript, voice and text chats from various internal sources along with publicly available social media data sources. We use unigram, bigram, trigram and n-grams textual features in terms of multimodal sentiment analysis. Our model's training dataset includes customer sarcastic reviews in the context of customer sentiment. In this experiment polarity, negation and sarcasm are considered and classified as positive and negative sentiment.

Our main goal is to build a state-of-the-art learning model that utilizes proven binary and multi-class algorithms, methods, and word embedding techniques for classification. The following below are contributions made towards that goal:

- Solving issues with context-based sentiments through multi-layer SA model with Bi-LSTM, *fast*Text and LSVM. Data pre-processing is customized to fit the model requirement for customer review and speech transcript dataset.
- Solving vocabulary issues with predictions made from datasets containing mixed language texts by adding input and service layers to detect the baseline language, translates them into English, and form a transcript.
- Transcript and Translate service layers are added to model input layer for train and testing the domain-base mixed-language dataset to avoid out of vocabulary (OOV) issues.
- Saving cost by providing new data pipeline techniques while having increased performance to train models with large datasets.
- LSVM, *fast*Text and SA-BLSTM models hyperparameters are fine-tuned based on dataset.

This paper details several contributions from various researchers, such as relating literature, published works, and research papers and are taken into review. The methods used, as well as the pre-processing steps and flow of data are presented. This paper also describes the approach taken for the ML and DL models, as well as the architecture of various algorithms used for the model. Details of the experiment are documented throughout, such as the setup and the results. In the end, the results of the concluding model are shown, as well as a proposal for the future.

# RELATED WORKS

Reviewed recent research papers and researchers' contributions towards text classification, sentiment prediction. Our review focused on supervised ML, DL and SA research papers.

*Kruspe et al. (2020)* published a paper related to COVID-19 cross-language SA of European Twitter messages from Italy, Spain, France, and Germany. Neural network model used with the pre-trained word or sentence embedding. A model was constructed with a fully-connected Rectified Linear Unit (ReLu) layer to process output from embedded vector and a regression output layer with sigmoid activation. In this experiment the following pre-trained word embeddings models used: a skip-gram version of word2vec (trained English-language Wikipedia data), and a multilingual version of BERT (trained on Wikipedia data and 160 million COVID-19 tweets key-words). The result showed, based on analysis, 4.6 million tweets in which 79,000 tweets contain one keyword of COVID-19. In this paper, researchers took geolocation-based data, and trends were varying; this study will be continued to collect tweet data, from other countries and compare the results, and also move from the binary sentiment scale to a more complex model. *Kumar et al., (2021)* published an article related to a machine learning scraping tool for a data fusion in the analysis of sentiments about supporting business decisions with human-centric AI (HAI) explanations. The multinomial Naïve Bayes (NB, k-nearest neighbours (KNN), SVMs and multinomial Bayesian classifiers are used for sentiments analysis. This study results revealed KNN outperformed other models.

*Gaye, Zhang & Wulamu (2021)* published an article related to employee sentiment using employees' reviews. This study used traditional classifiers and vector stochastic gradient descent classifier (RV-SGDC) for sentiment classification. RV-SGDC is a combination of Logistic Regression (LR), Support Vector Machines (SVM), and Stochastic Gradient Descent (SGD) model. The study result showed RV-SGDC outperforms with a 0.97% accuracy compare to other models due to its hybrid architecture.

*Chinatalapudi, Battineni & Amenta (2021)* published an article paper related to SA of COVID-19 tweets using Deep Learning (DL) models. These researchers' motto behind this study to analyze tweets by Indian netizens during the lockdown, collected tweets between 23 March 2020 and 15 July 2020 and text has been labelled as fear, sad, anger and joy, analysed data using the new deep learning model name called Bi-Directional Encoder Representation from Transformers (BERT). The BERT model result was compared with traditional Logistic Regression (LR), Support Vector Machines (SVM), Long Short-term Memory (LSTM). The BERT model result shows more accuracy 89%, compare to other model's accuracy LR 75%, SVM 74.75% and LSTM 65%. This experiment classified sentiment into fear, sad, anger and joy based on the key-words.

Most recently, *Alharbi et al. (2021)* published a research article related to evaluation of SA using the Amazon Online Reviews dataset. Researchers evaluated different deep learning approaches to accurately predict the customer sentiment, categorized as positive, negative and neutral. The variation of simple Recurrent Neural Network (RNN) such as Long Short-Term Memory Networks (LRNN), Group Long Short-Term Memory Networks (GLRNN), Gated Recurrent Unit (GRNN) and Updated Recurrent Unit (UGRNN).

for Amazon Online Reviews. All evaluated RNN algothims were combined with word embedding as feature extraction approach for SA including the following three methods Glove, Word2Vec and fastText by Skip-grams. A combination of five RNN variants with three feature extraction methode was evaluated; the evaluation result was measured based on accuracy, recall, precision and F1 score. It was found that the GLRNN with fastText feature extraction scored the highest accuracy of 93.75%. Researchers try to solve programming problem for beginners to code and find next word, used conventional LSTM model with word embedding, dropout layer with an attention mechanism. This model result showed the pointer mixture model succeeded in predicting both the next within-vocabulary word and the referenceable identifier with higher accuracy than the conventional neural language model alone in both statically and dynamically typed languages.

*Labhsetwar (2020)* published a paper related to customer churn prediction, the traditional Logistic Regression (LR), Gaussian Naïve Bayes (GNB), Adaptive Boosting (AdaBoost), Extra Gradient Boosting (XGB), Stochastic Gradient Descent (SGD), Extra Trees and SVM classifiers are used for this experiment. The results showed Extra Trees classifier outperformed, SVM and XGB classifier performed well for Telecom (UCI repository) dataset. Many researchers (*Ikonomakis, Kotsiantis & Tampakas, 2005*) have shown combining multiple classifiers improve performance and classification accuracy of model in the context of combining multiple classifiers for text categorization.

*Gaye, Zhang & Wulamu (2021)* published an article related to employee sentiment using employees' reviews using the traditional classifiers and vector stochastic gradient descent classifier (RV-SGDC) for sentiment classification. RV-SGDC is a combination of Logistic Regression (LR), Support Vector Machines (SVM), and Stochastic Gradient Descent (SGD). The result showed RV-SGDC outperforms with a 0.97% accuracy compare to other models due to its hybrid architecture.

*Kumar & Chinnalagu (2020)* presented a research study paper related to sentiment and emotion in social media COVID-19 conversations. This research used variants of RNN algorithms and evaluated a multi-class neural network model using Bi-directional Long Short-term memory (Bi-LSTM) with additional layers to process the COVID-19 long text social media posting, overcome model outfitting, accuracy and performance problems. The experimental result showed SAB-LSTM model outperformed the traditional LSTM, Bi-LSTM models and sentiment prediction was context-based. Authors planned to extend their model for future research with domain-based dataset for customer SA problems, comparing with other models to improve the prediction accuracy and performance.

*Kowsari et al. (2019)* presented a text classification survey paper. In this paper, researchers discussed about existing classification algorithms, feature extraction, dimensionality reduction, and model evaluation methods, and also addressed critical limitation of each one of these components of the text classification pipeline. The following components are discussed: algorithms such as Rocchio, bagging and boosting, logistic regression (LR), Naïve Bayes Classifier (NBC), k-nearest Neighbor (KNN), Support Vector Machine (SVM), decision tree classifier (DTC), random forest, conditional random field (CRF), and deep learning. Feature's extraction methods such as Term Frequency-Inverse document frequency (TF-IDF), term frequency (TF), and word-embedding

methods such as Word2Vec, contextualized word representations, Global Vectors for Word Representation (GloVe), and fastText. Dimensionality reduction methods such as Principal component analysis (PCA), linear discriminant analysis (LDA), non-negative matrix factorization (NMF), random projection, Autoencoder, and t-distributed Stochastic Neighbor Embedding (t-SNE). Evaluation methods such as Accuracy, Fβ, Matthew correlation coefficient (MCC), receiver operating characteristics (ROC), and area under curve (AUC).

*Joulin et al. (2016)* at Facebook's AI research lab (FAIR) released and presented a linear text classifier fastText library a paper. It proved that fastText library can be transformed into a simpler equivalent classifier, and also proved that the necessary, sufficient dimensionality of the word vector embedding space is exactly the number of document classes. Experiment results show that combination of bag of words and linear classification methods fastText accuracy is same or slightly lower than deep learning algorithms, fastText performs well in normal environment setup, even without using high performance GPU servers.

*Kowsari et al. (2017)* employed deep learning methods to multi-class documents classifications. The traditional multi-class classification works well for a limited number classes and the performance drops when increasing the number of classes and documents. To solve the performance problems, experimented combination of deep learning, recurrent and convolutional neural network models. This combined neutral networks, hierarchical DL classification model (HiDLTex) result showed more accuracy than traditional SVM and Naïve bayes models.

*Qu, Ifrim & Weikum (2010)* presented a paper at the International Conference on computational Linguistic related to Bag-of-Opinions method for review rating prediction from sparse text patterns. Customers are writing their comments with implicitly expressing their opinion polarities as positive, negative, and neutral, and also providing numeric ratings of products. The numerical review rating prediction is harder than classifying by polarity. In this paper discussed about a unigram-based regression model each unigram gets a weight indicating its polarity and strength rating, for *e.g.*, "This product is not very good" Vs "This product is not so bad", in this *e.g.*, unigram regression model consider weight to "good" as positive and "bad" as negative, and it assigns the strong negative weight to "not", combining this weight, it was not predicted the true intention of opinion phrases. These models are not robust and referred unigram regression model as polarity incoherence. To overcome these two models, introduced a novel kind of Bag-of-opinion (BoO) with approach of cumulative linear offset (CLO) model representation, where an opinion, within a review consists of the following three components, a root word, a set of modifier words from the same sentence, and one or more negation words. For a phrase *e.g.*, "not very helpful" has opinion root word "helpful", modifier word "very" and a negation word "not". Enforced polarity coherence by the design of a learnable function that assigns a score to an opinion by ridge regression, from a large, domain-independent corpus of reviews. All Amazon reviews datasets used for BoO model training and testing regardless of domains.

*Zhang & LeCun (2016)* published a paper to determining the explicit or implicit meaning of words, phrases, sentences and paragraphs, and making inferences about

these properties such as words and sentence of these texts has been traditionally difficult because of the extreme variability in language formation. The text understanding is another area of research to understand the text formed in natural languages such as English, Chinese, Spanish and others. To solve text understanding problem convolutional networks (ConvNet) models were used for research studies. For English text understanding, model was built using these 70 characters, including 26 English letters, 10 digits, new line and 33 other characters.

## Methodologies and process flow

Recently, researchers are used deep learning and neural network models for SA problems, however neural network approach cost more compare to traditional baseline methods for both supervised and unsupervised learning. The combination of right methodologies and text classification algorithms are contributed to overcome SA models accuracy and performance problems. The following process flow diagrams in Fig. 1 show the steps followed for this experiment. In general, the data pre-processing step, data obtain from publicly available customers review data is very often incomplete, inconsistent and filled with a lot of noise and it is likely contained errors not suitable for training and testing the machine learning models.

The following minimal syntactical data preprocessing steps of lowercasing all words, removing new lines, punctuation, special characters and stripping recurring headers are needed for neural networks and embedding models. To improve data quality, introduced additional steps which includes stop words removal, text standardization, spelling correction, correcting the negation words, tokenization, stemming, and Exploratory Data Analysis (EDA). Figure 2 shows the three proposed models input and output data flow. These models are customized to fit the dataset. In this experiment, added additional layers SA-BLSTM to handle large volume of customer reviews and speech transcript data. *fast*Text and SVM are linear models. All these pre-trained models and developed algorithms can be used for production purpose on real-time SA business applications.

# PROPOSED MODELS

LSVM, *fast*Text and SAB-LSTM models are used for this experiment, before building the models, reviewed algorithms towards solving the short and long text classification and SA problems. In these following sections, explained all these three custom model architectures.

## Support Vector Machine (SVM)

SVM is used for text classification problems, this algorithm is viewed as a kernel machine, and the kernel functions can be changes based on the problem. Definition of Support Vector Machines (SVM): It performs classification by finding the hyperplane that maximize the margin between the two classes. The support vector is the vectors that define the hyperplane. For instance: given pictures of apples and oranges, state whether the object in question is an apple or an orange. Equally well, it can predict whether a customer is satisfied or not satisfied given customers positive and negative sentiment data. SVM performs classification by finding the hyperplane that maximize the margin between the two classes.

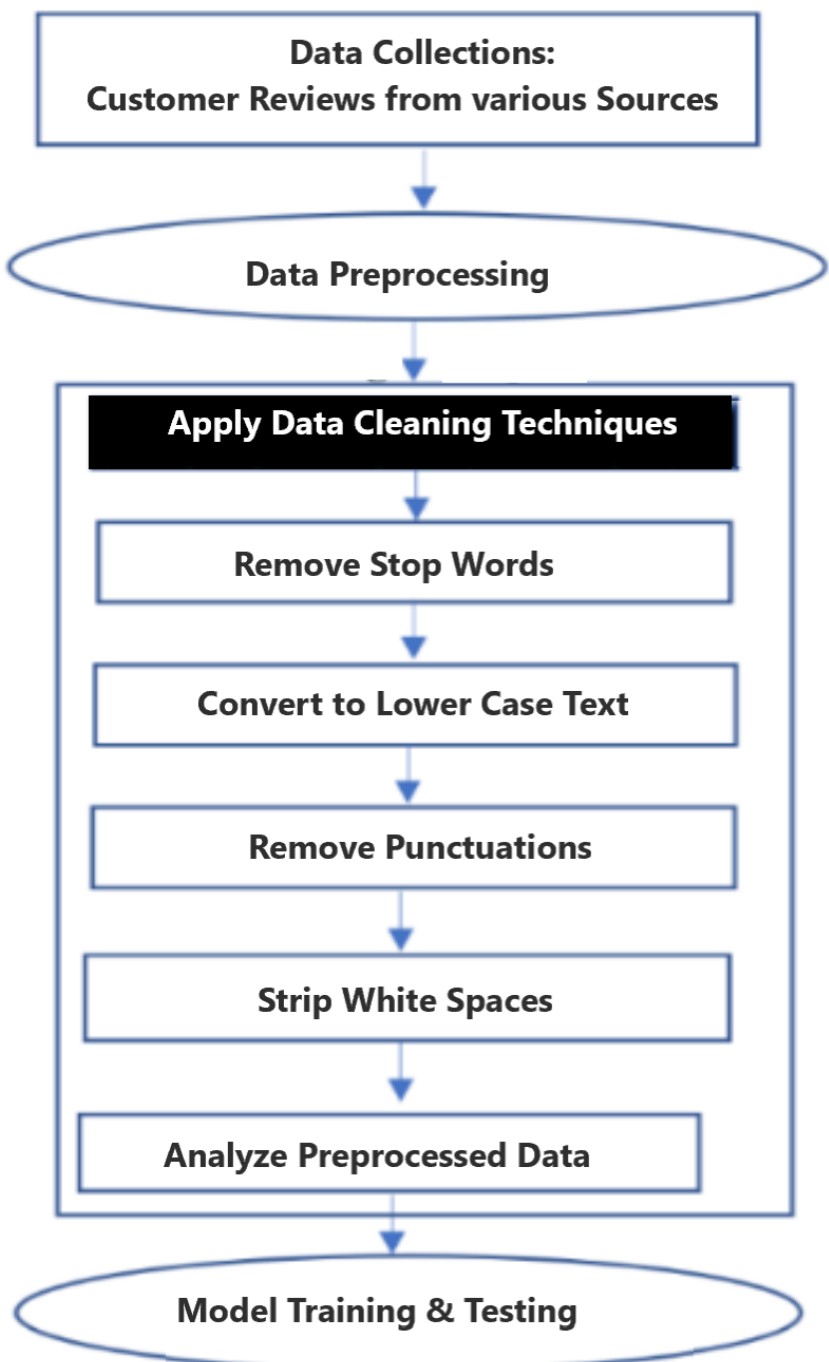

**Figure 1** Methodology and process flow diagram.

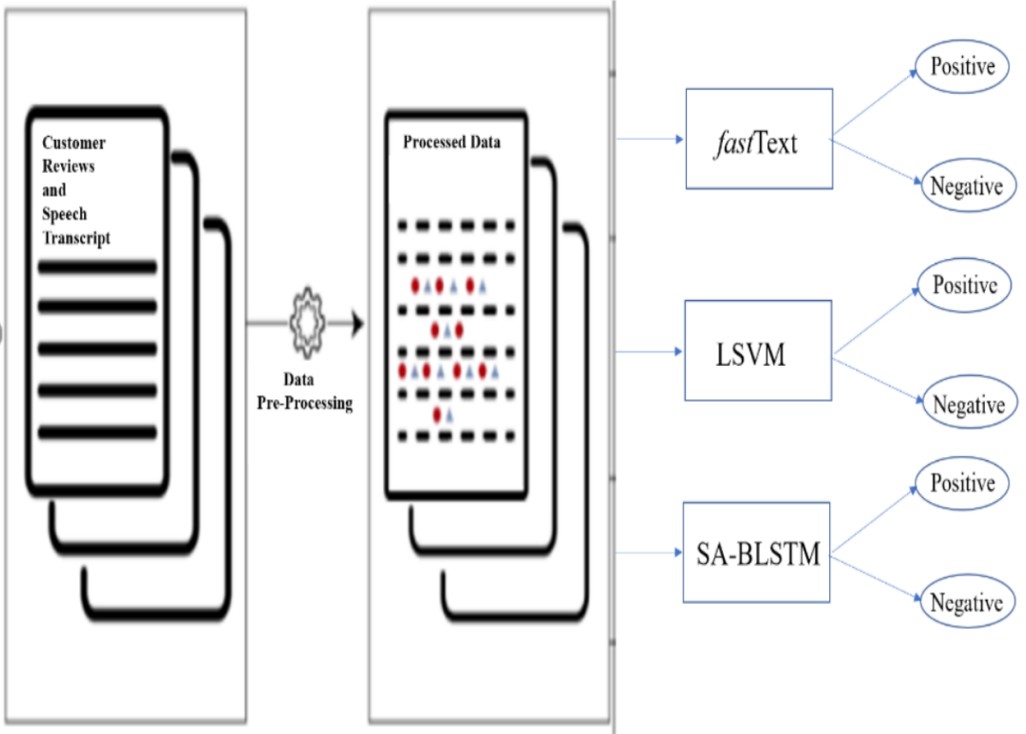

**Figure 2** Input and model output data flow.

In other words, SVM is that the partition which segregate the classes. Figure 3 shows an example and definition of a point and vector, plotted a point $A(4,2)$ and any point,

$$x = (x_1, x_2), x \neq 0 \tag{1}$$

A vector is an object that has both a magnitude and a direction. In geometrical term, a hyperplane is a subspace whose dimension is one less than that of its ambient space. If a space is in 3-dimensional then it's a plane, if a space is in two dimensions, then it's a line, if the space in one dimension, then it's a point. Figure. 4 shows the hyperplane definition.

The linear and non-linear classifier (*Kowsari et al., 2019*) data separation shown in the Fig. 5 for the 2-dimensional dataset. If the dataset is separable then linear kernel works well for classification. Because of the following reasons, the linear kernel is used for the text classification. The most text categorization problems are linearly separable and the linear kernel works well with lot of features and also less parameters to *Joachims (1998)* optimize for training the model.

Figure. 6 shows the linear kernel (*Crone, Lessmann & Stahlbock*) SVM model. There many kernel functions have been developed over the years; a kernel is a function, that

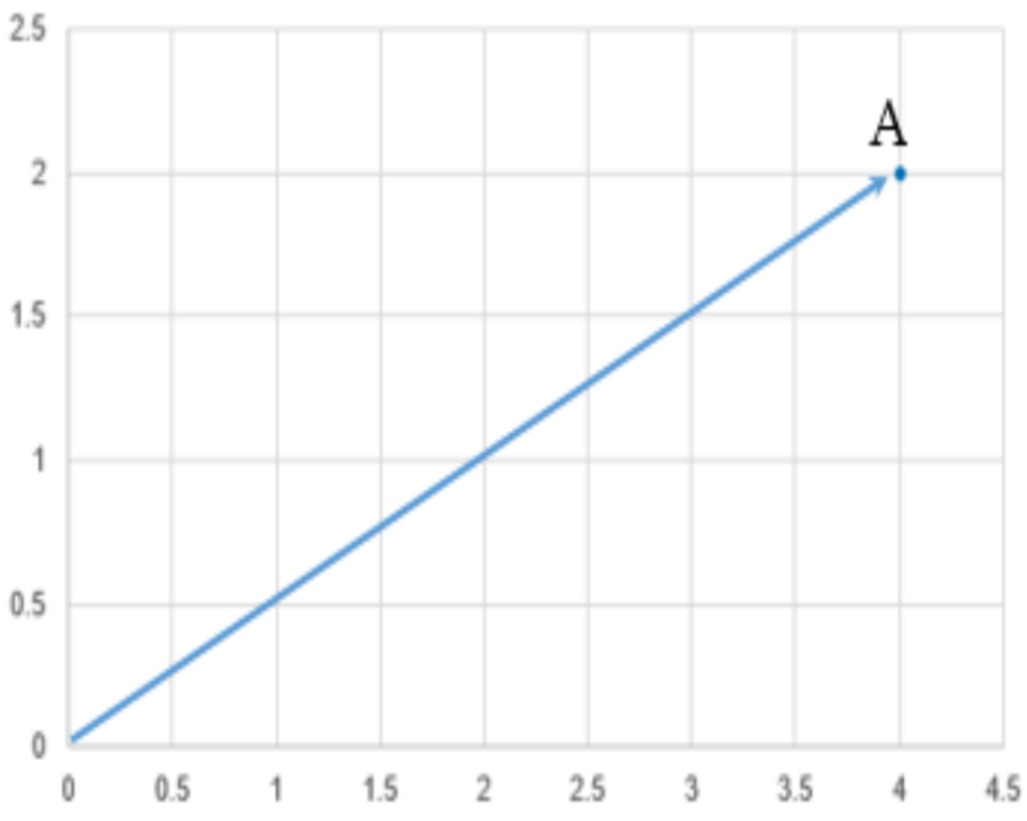

**Figure 3**  Representation of vector.

returns the result of a dot product performed (*Alexandre Kowalczyk, 2017*) in another space.

The linear kernel is simplest kernel function $k(x,y)$, it is given by the inner product $<x,y>$, and an optional constant c.

$$k(x,y) = x^T + c. \tag{2}$$

Linear kernel is used in this experiment for text classification. The Polynomial kernel, RBF kernel and String kernel functions can be used for other classifications problems.

## fastText

Facebook AI research (FAIR) lab release an open-source free library called fastText for text representation and classification. It's a lightweight method and work on standard generic hardware with multicore CPU. *fast*Text approach evaluated for tag prediction and sentiment analysis by FAIR. *fast*Text experiments (*Joulin et al., 2016*) show that it is often on par with recently proposed DL methods in terms of accuracy, performance, faster for training and evaluation. Facebook allows research community to build the models on top of the fastText open-source code. fastText introduce a new word embedding approach an extension of the continuous skipgram and Continuous Bag of Words (CBOW) model like word2vec, where each word is represented as a bag of character $n$-grams. The original

Define the hyperplanes $H$ such that:
$w \bullet x_i + b \geq +1$ when $y_i = +1$
$w \bullet x_i + b \leq -1$ when $y_i = -1$

$H_1$ and $H_2$ are the planes:
$H_1$: $w \bullet x_i + b = +1$
$H_2$: $w \bullet x_i + b = -1$
The points on the planes $H_1$ and $H_2$ are the tips of the Support Vectors
The plane $H_0$ is the median in between, where $w \bullet x_i + b = 0$

d+ = the shortest distance to the closest positive point
d- = the shortest distance to the closest negative point
The margin (gutter) of a separating hyperplane is d+ + d–.

**Figure 4** Hyperplane.

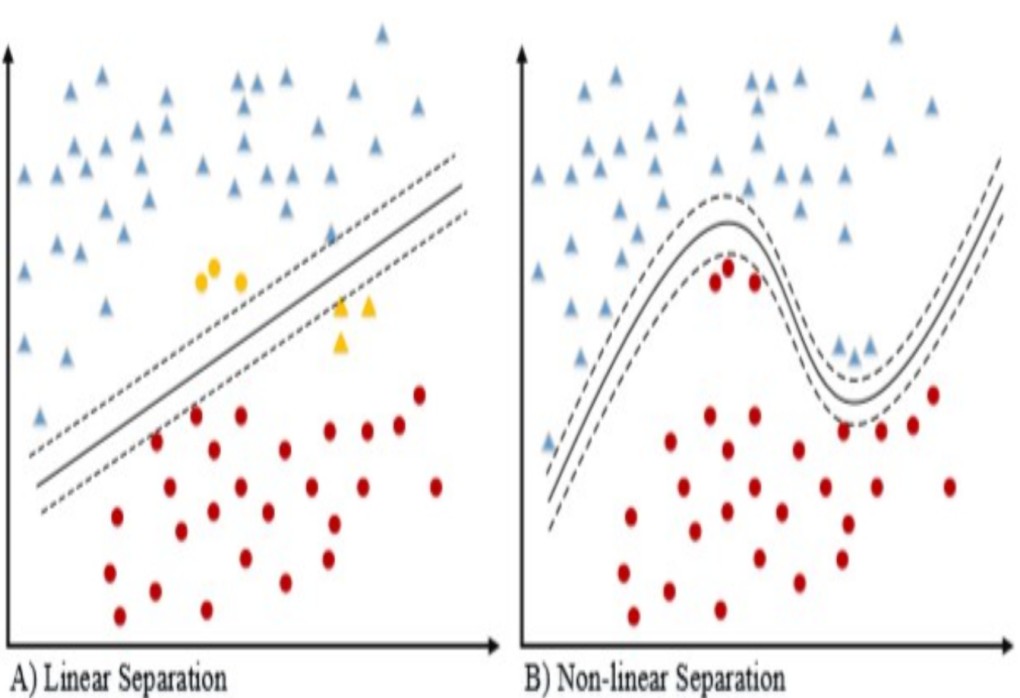

A) Linear Separation          B) Non-linear Separation

**Figure 5** Data separation linear and non-linear.

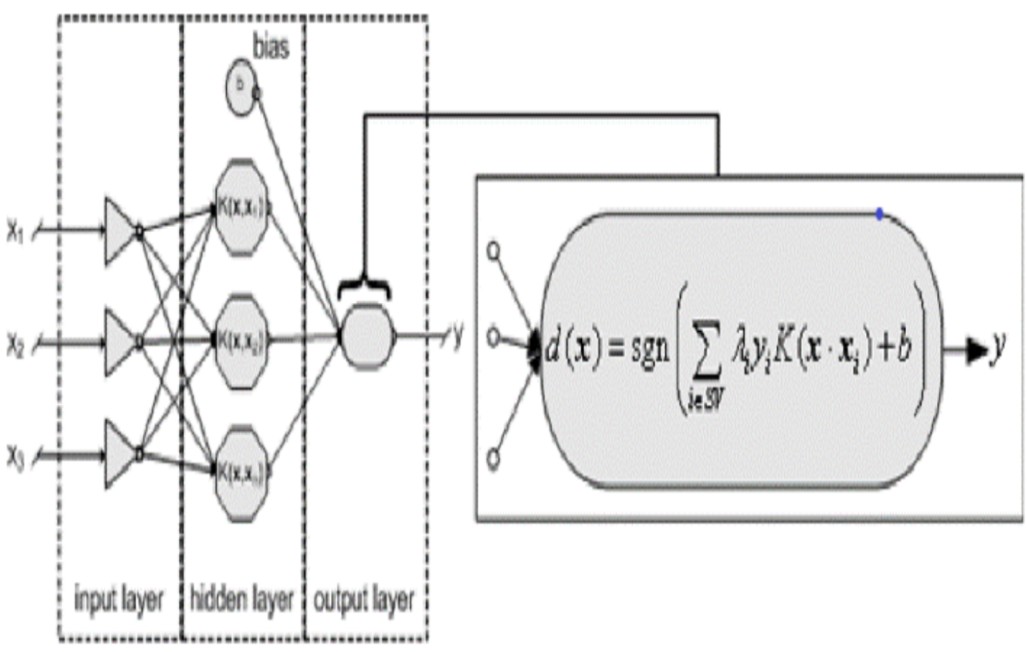

**Figure 6** Linear support vector machine.

version of *fast*Text is trained on (*Nitsche & Halbritter 2019*) Wikipedia and its available for 294 languages. The main difference between word2vec and *fast*Text is that *fast*Text sees words as the sum of their character n-grams and it treats a vector representation is associated to each character *n*-gram and words being represented (*Bojanowski et al., 2017*) as the sum of these representations. This new approach has clear advantages, as it can calculate embeddings even for out-of-vocabulary (OOV) words. However, word2vec embedding approach treat word as the minimal entity and try to learn their respective embedding vector, in case if the word does not appear in the training corpus, then it fails to get word vector representation. Figure 7 shows the CBOW and Skip-gram model architecture, The CBOW is the distributed representation of context model, it predict the words in middle of a sentence based on surrounding words. However, the Skip-gram predicts context within a sentence (*Mikolov, Le & Sutskever*).

The Skip-gram maximize the average log probability for input of training words $w$,

$$w_1, w_2, w_3 \ldots, w_T. \tag{3}$$

Eq. (4) is used for computing probability.

$$\frac{1}{T} \sum_{t=1}^{T} \sum_{j=-k}^{T} \log p(w_{t+j}|w_t). \tag{4}$$

Here k represents the size of the training window and function for $w_t$ word in the middle.

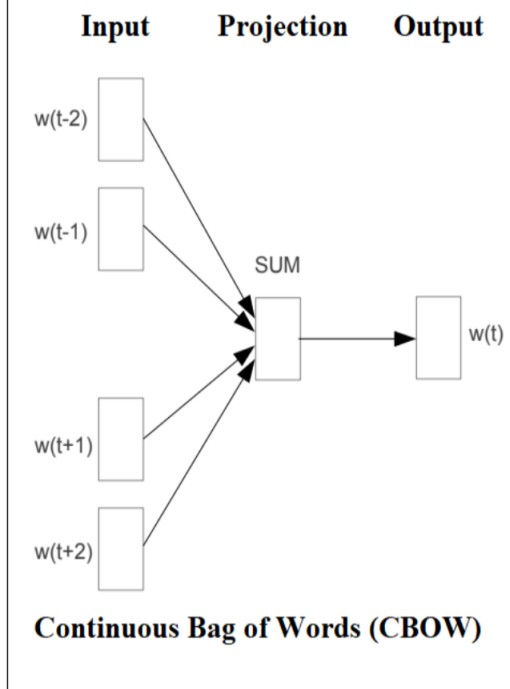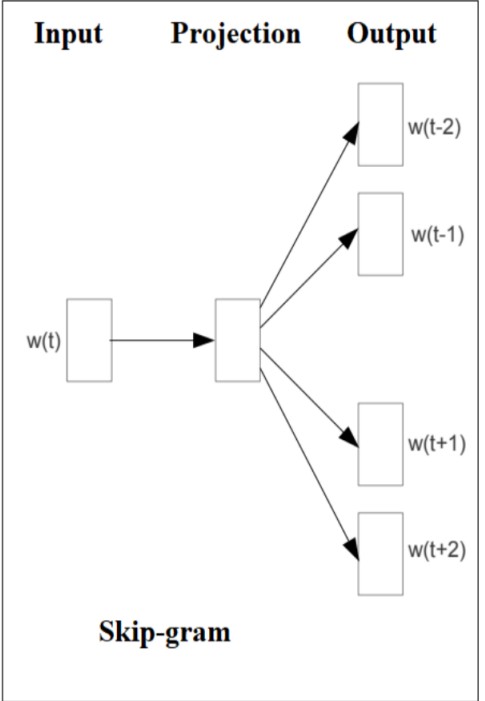

**Figure 7  Graphical representation of the CBOW and Skip-gram model.**

−k to k is the representation inner summation and it computes the log probability of the word $w_{t+j}$ prediction for word in the middle $w_t$. The outer summation words are based on the corpus used for model training.

The skip-gram model consists of input $u_w$, and output $v_w$, vectors associated with each word $w$. The following probability (Eq. 5) is used to predict the word $u_i$, from $w_j$.

$$p(w_i|w_j) = \frac{\exp(u_{w_i}\top v_{w_j})1}{\sum_{l=1}^{V}\exp(u_{w_l}\top v_{w_j})}. \tag{5}$$

Here, the total number $V$ of words in the given vocabulary.

CBOW and skip-gram models are obtaining the semantic information during the large datasets used for training. The words which are closely related has the similar vector representation the words. For example, school, college, university, education words are having similar context, similarly orange and apple are having similar context representations.

The Fig. 8 shows the architecture (*Joulin et al., 2016*) of a simple linear model of *fast* Text with $N$ gram features.

$$x_1, x_2, \ldots .x_N. \tag{6}$$

The features are embedded and averaged to form the hidden variable. This model is a simple neural network with only one layer. The bag-of-words representation of the text is

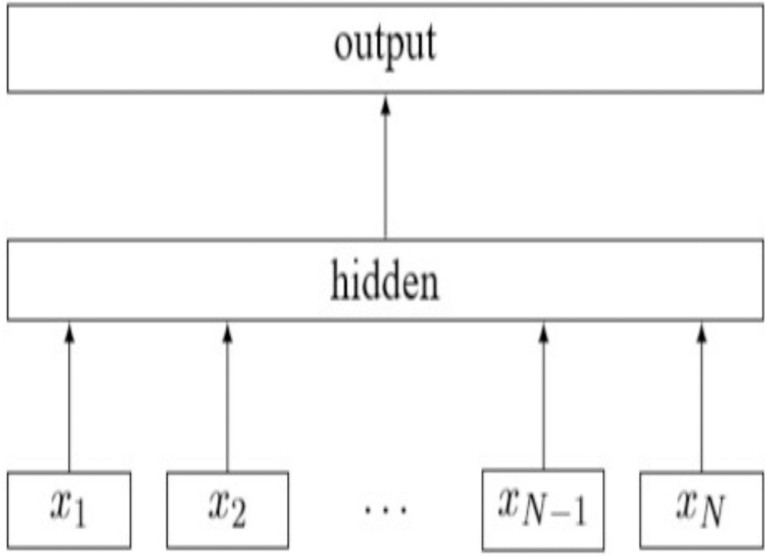

**Figure 8** **A simple linear fastText architecture.**

first fed into a lookup layer, where the embedding is fetched for every single word. Then, those word embedding are averaged, so as to obtain a single averaged embedding for the whole text. At the hidden layer we end up with n words $x$ dim number of parameters, where dimension is the size of the embedding and n words is the vocabulary size. After the averaging, we only have a single vector which is then fed to a linear classifier: we apply the softmax over a linear transformation of the output of the input layer. The linear transformation is a matrix with dimension1 $x_N$ output, where N output is the number output classes (*Mestre, 2018*). The following Eq. (7) is the negative log likelihood function of fastText model.

$$-\frac{1}{N} = \sum_{n=1}^{N} y_n \log(f(BAx_n)).$$  (7)

Here, $x_n$ represents the n-gram feature of the word,

A represents the lookup matrix of the word embedding,

B represents the linear output of the model transformation,

$f$ represents the softmax function.

The softmax function calculates the probabilities distribution of the event over n different events. The softmax takes a class of values and converts them to probabilities with sum 1. So, it is effectively squashing a k-dimensional vector of arbitrary real values to k-dimensional vector of real values within the range 0 to 1. Eq. (8) is the softmax function $f$ of fastText.

$$\text{softmax}(z) = \frac{\exp(z)}{\sum_{k=1}^{K} \exp(z)}.$$  (8)

The fastText has the following tuning parameters: *Epoch:* By default, the model is trained on each example for 5 epochs, to increase this parameter for better training specify the

number of epoch argument. *Learning rate (lr):* The learning rate controls how "fast" the model updates during training. This parameter controls the size of the update that is applied to the parameters of the models. Changing learning rate implies changing the learning speed of our model is to increase (or decrease) the learning rate of the algorithm. This corresponds to how much the model changes after processing each example. A learning rate of 0 would means that the model does not change at all, and thus, does not learn anything (*Mestre, 2018*). Note that this calculation of the best model is going to be quite expensive. There is no magic formula to find the hyperparameters for the best model. Just taking one hyperparameter, the learning rate, would make the calculation impractical. This is a continuous variable and it would need to feed in each specific value, compute the model, and check the performance. *Loss function:* In this we are using softmax as loss function. The most popular methods for learning parameters of a model are using gradient descent. Gradient descent is basically an optimization algorithm that is meant for minimizing a function, based on which way the negative gradient points toward. In machine learning, the input function that gradient descent acts on is a loss function that is decided for the model. The idea is that if we move towards minimizing the loss function, the actual model will "learn" the ideal parameters and will ideally generalize to out-of-sample or new data to a large extent as well. In practice, it has been seen this is generally the case and stochastic gradient, which is a variant of gradient descent, has a fast-training time as well. Since it needs to obtain the posterior distribution of words, the problem statement is more of a multinomial distribution instead of a binary.

Figure 9 shows the text documents process flow using fastText linear model. In this model text classification pipe-line, raw text documents are processed using data pre-processing steps described in Fig. 1 and processed text data tested using fastText model, the model output classified the output into two classes (Satisfied and Not-Satisfied). *fast*Text is a classification algorithm and C++ used to compile fastText mode. It provides high accuracy as well as good performance (*Zolotov & Kung, 2017*) during training and testing the model.

## SA-BLSTM model

SA-BLSTM is a sequence processing model (*Kumar & Chinnalagu, 2020*). The Bidirectional LSTM is used in this model. The extended LSTM architecture is shown in Fig. 10 with Input, Output, multiplicative Forget gates and all the gates are using (*Mikolov et al., 2013*) activation function $f$ sigmoid.

The Constant Error Carousels (CEC) is the central feature of LSTM (*Kumar & Chinnalagu, 2020*) and it solves the vanishing problem. CECs back flow is constant when there are no input or error signals to the cell. Input and output gates protect CECs error flow from forward and backward activation. If the gates are closed or the gates activation is around 0 then the irrelevant input will not enter the cell. The Forward pass LSTM computes the output of the network given the input data, the Backward pass LSTM computes the output error with respect to the expected output and then go backward into the network and update the weights using gradient descent. To compute (Tomas et al., 2013) the network weight for a single input to output network, the back propagation (BP)

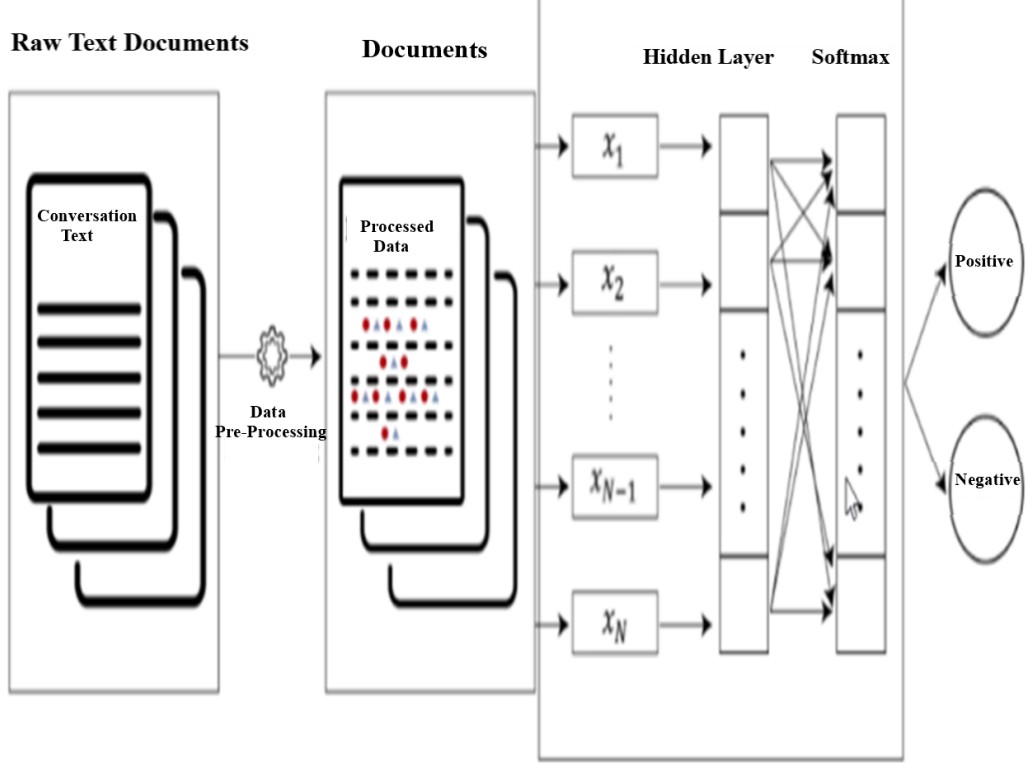

**Figure 9  fastText model text document process flow.**

uses the loss function to compute the gradient. The following are the equations for gates and activation functions. LSTM Gates are the activation of sigmoid function, between 0 and 1 is output value of the sigmoid. When the gates are blocked the value is 0 and when the value is 1 then gates allow the input to pass through.

$$\text{sig}(t) = \frac{1}{1+e^{-t}}. \tag{9}$$

Here are the equations for all three gates.
Input Gate

$$i_t = \sigma(\omega_i[h_{t-1}, x_t] + b_i). \tag{10}$$

Output Gate

$$o_t = \sigma(\omega_f[h_{t-1}, x_t] + b_o). \tag{11}$$

Forget Gate

$$f_t = \sigma(\omega_o[h_{t-1}, x_t] + b_f). \tag{12}$$

$\sigma$ Represents sigmoid function, $x_t$ Represents input at current timestamp. $h_{t-1}$ Represents LSTM block output of previous state at timestamp t $-1$. $\omega_i$, $\omega_f$, and $\omega_o$ are represents weight for the input, forget and output gates. $b_i$, $b_o$ and $b_f$ are represents bias

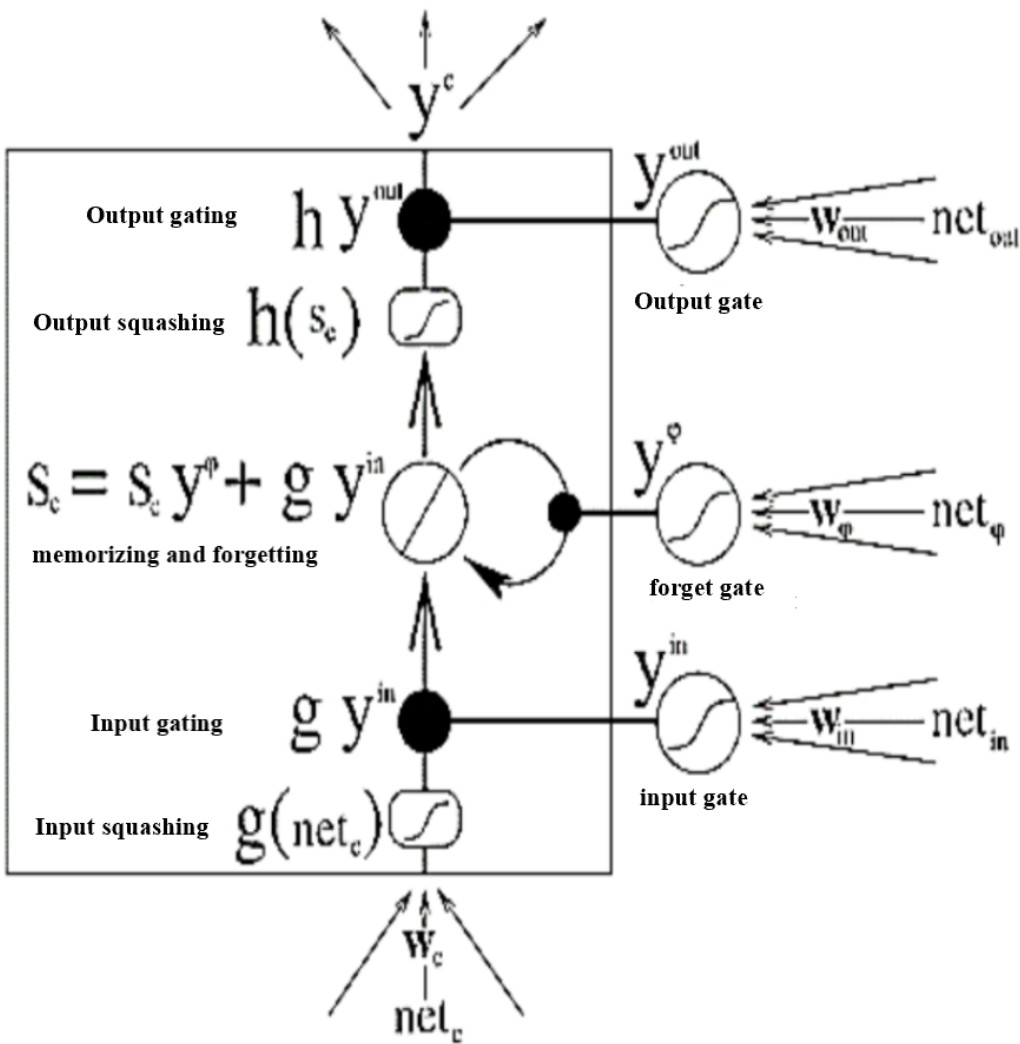

**Figure 10 Extended LSTM with multiplicative Forget gate.**

for input, output, forget gates. The following are equations for cell state for gates (*Kumar & Chinnalagu, 2020*).

Cell State of Input gate

$$\tilde{c}_t = tanh(c\omega\left[h_{t-1}, x_t\right] + b_c).\tag{13}$$

Cell Sate of Output gate

$$c_t = f_t * c_{t-1} + i_t * \tilde{c}_t.\tag{14}$$

Cell Sate of Forget gate

$$h_t = o_t * tanh(c^t).\tag{15}$$

$\tilde{c}_t$ Represents input gate cell state at timestamp $(t)$. $c_t$ Represents memory cell state at timestamp $(t)$. $h_t$ Represents cell state of final output at timestamp $(t)$.

This model (*Kumar & Chinnalagu, 2020*) consists of Input, Language detection and translation, embedded, Bi-Directional LSTM neutral network layer, dropout, dense and output layers. The input layers process the multilingual mixed customer reviews and speech transcript dataset and vectorizing the data using word embedding technique, each post has one or more sentences, and each sentence is composed with $n$ number of words sequence. Here $x$ representing input during the language detection, language translation process.

$$x = x_1, x_2, x_3, \ldots . x_T. \tag{16}$$

In the detection layer, input text processed to detect the non-English text, here $d$ represents input to detection layer.

$$d = d_1, d_2, d_3, \ldots . d_T. \tag{17}$$

If the input text identified as non-English text, then the language translation layer converts the text to English, here t represents the input of translation layer.

$$t = t_1, t_2, t_3, \ldots . t_T. \tag{18}$$

The output of the translation layer processed by embedding layer, each input word converted to vector, here S represents vector value.

$$S = w_1, w_2, w_3, \ldots . w_n. \tag{19}$$

The Bi-Directional LSTM (BLSTM) is a sequence processing model, it consists of two LSTM units, (*Gopalakrishnan & Salem, 2020*) one unit taking the input in a forward direction and other unit taking the input in a backward direction. It effectively processes the input and context available to the network. Figure 11 shows the mixed-language data processing flow, the language detection and translation layers convert the non-English to English and then it's embedding the words.

Input layer fed the embedded dataset to Bi-LSTM model and it processes vector output of the embedded layer. The following Fig. 12 shows the SA-BLSTM model architecture. This model used for both binary and multiclass classifications and SA applications.

## EXPERIMENT AND RESULTS

For models training and testing, used Windows 64-bit Operating System with Intel core i7 processor, 16 GB Memory, and on-board GPU NVIDIA MX150 server environment. Developed models using Python, Jupyter, Anaconda IDE and used Python libraries, Pandas for data load processes, Numpy for mathematical operations, Seaborn and Matplotlib for plotting graph, NLTK Tool kit, Wordcloud and Sklearn used for build, train and test the models. This novel method uses a memory caching technique with automated custom Python scripts to speed-up data preprocessing tasks during model training on large volume dataset.

To compare various traditional models results for Twitter dataset, we collected related (*Mittal & Patidar, 2019*) experimental results. Table 1 shows the various models accuracy. Figure 13 shows the experiment result of fastText model with bigram.

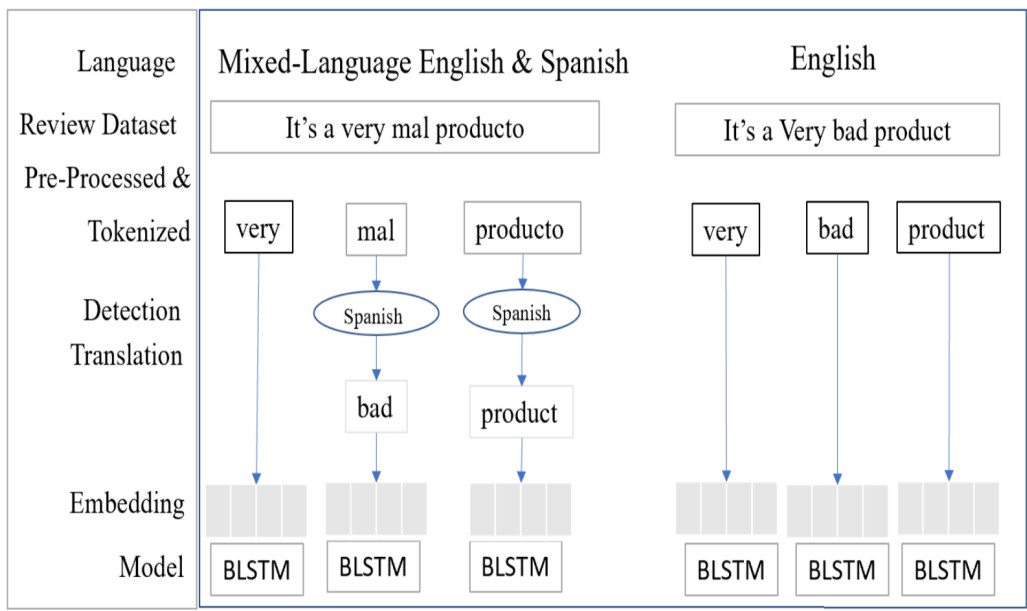

**Figure 11  SA-BLSTM mixed-language text data process flow.**

**Table 1  Comparison of related works.**

| Et.al | Dataset | Model | Accuracy % |
|---|---|---|---|
| Geetika Gautam | Twitter Customer Review | SVM | 85.5 |
| | | Max Entropy | 83.8 |
| | | Naïve Bayes | 88.2 |
| | | Semantic Analysis (WordNet) | 89.9 |
| Seyed-Ali Bahrainaian | Twitter data on smart phones | Hybrid Approach- Unigram, Naive Bayes, MaxEnt, SVM | 89.78 |
| Neethu M.S | Twitter data on electronic products | Naïve Bayes | 89.5 |
| | | SVM | 90 |
| | | Max Entropy | 90 |
| | | Essemble | 90 |
| Dhiraj Gurkhe | Twitter Data | Unigram | 81.2 |
| | | Bigram | 15 |
| | | Unigram-Bigram | 67.5 |

The dataset was collected from publicly available Twitter information, IMDB movie reviews, Amazon product review and Yelp sentiment analysis data source from kaggle.com, for a total of 778,631 datasets, 70% (545041) of data used for training and 30% (233590) of data used to test the models which includes Kaggle.com sentiment datasets, chat conversations from chat application and transcript of sample audio files. These data sources details and URLs are listed in Table 2.

Designed and developed fastText and Linear SVM (LSVM) models, used the linear kernel setting for LSVM model and for fastText model used the epoch $=10$, lr $=0.01$ and loss $=$softmax parameters. Both the models trained with unigram($n=1$), bigram($n=2$)

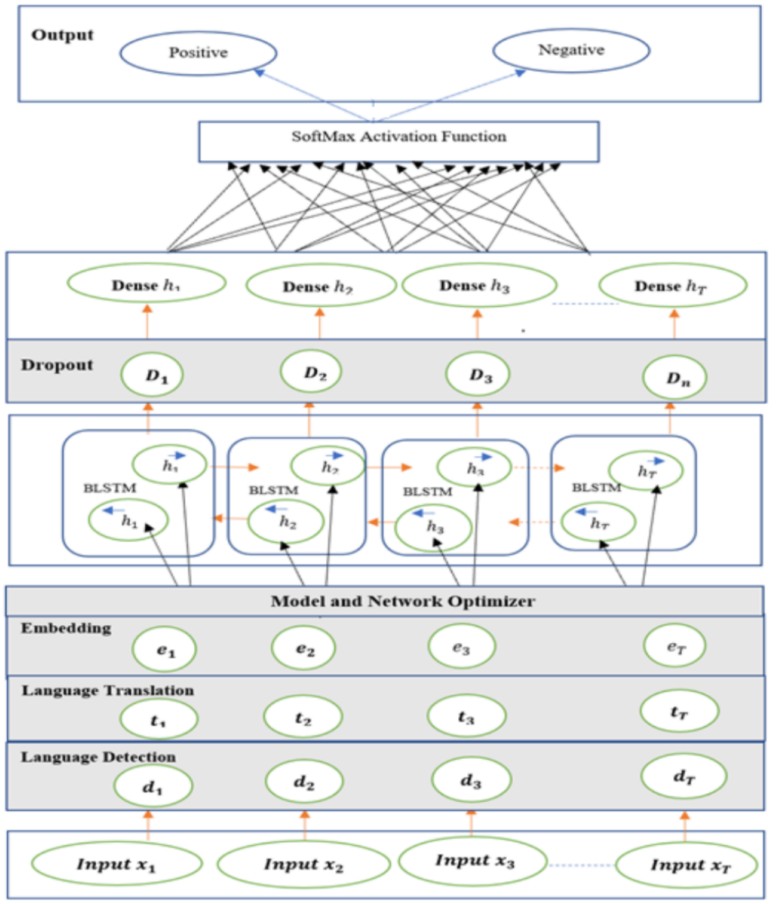

**Figure 12  Sentiment analysis bi-directional LSTM model (SA–BLSTM).**

and trigram($n = 3$) and n-grams parameters. The unigram ($n = 1$) model result shows the polarity incoherence (*Qu, Ifrim & Weikum, 2010*) due to unigram model gets a weight indicating its polarity and strength, for *e.g.*, *not so good Vs. not so bad*, the fundamental problem arises in unigram model when assign the weight to *not*. Analyzed the training result, 3-gram showed the better performance of LSVM and fastText model for this dataset. Tested with authors Pre-trained SBA-LSTM model using the same dataset. We also combined the datasets of Amazon, Yelp, Twitter, IMDB, Chat and audio transcripts for the LSVM and SA-BLSTM models.

During the model testing, captured the test results and tested n-gram features on both the models. The result shows for example, customer wrote the following review about phone purchase experience. *e.g.*, "*Even after three working days of phone purchase. Noticed that phone Service was not good.*"

Based on the unigram method the above text considers only one word for instance, in this case the above sentence "*phone Service was not good*", it can be written a

Probability(P) P("*phone Service was not good*")

| Model | AG | Sogou | DBP | Yelp P. | Amz. P. |
|---|---|---|---|---|---|
| BoW (Zhang et al., 2015) | 88.8 | 92.9 | 96.6 | 92.2 | 90.4 |
| ngrams (Zhang et al., 2015) | 92.0 | 97.1 | 98.6 | 95.6 | 92.0 |
| ngrams TFIDF (Zhang et al., 2015) | 92.4 | 97.2 | 98.7 | 95.4 | 91.5 |
| char-CNN (Zhang and LeCun, 2015) | 87.2 | 95.1 | 98.3 | 94.7 | 94.5 |
| char-CRNN (Xiao and Cho, 2016) | 91.4 | 95.2 | 98.6 | 94.5 | 94.1 |
| VDCNN (Conneau et al., 2016) | 91.3 | 96.8 | 98.7 | 95.7 | 95.7 |
| fastText, $h = 10$ | 91.5 | 93.9 | 98.1 | 93.8 | 91.2 |
| fastText, $h = 10$, bigram | 92.5 | 96.8 | 98.6 | 95.7 | 94.6 |

**Figure 13** **Comparison of related dataset.**

**Table 2** **Data Sources and URLs.**

| Sentiment analysis datasets | Data Source URL |
|---|---|
| Twitter | https://www.kaggle.com/c/twitter-sentiment-analysis/data |
| IMDB Movie Review | https://www.kaggle.com/columbine/imdb-dataset-sentiment-analysis-in-csv-format |
| Yelp | https://www.kaggle.com/omkarsabnis/sentiment-analysis-on-the-yelp-reviews-dataset/data |
| Amazon | https://www.kaggle.com/saurav9786/amazon-product-reviews |
| Chat | Our internal team chat conversations dataset |
| Audio Transcript | Our internal team audio recordings dataset |

$$= P(``phone") * P(``Service") * P(``was") * P(``not") * P(``good") \qquad (20)$$

From this equation here unigram $n = 1$ matches pattern word by word in this case "*good*" gets more weight considering it is an individual word so the SVM unigram predicts 32% as positive and fastText unigram predicts 19.37% as positive. So, to avoid this problem, used Bigram $n = 2$, where the algorithm considers the "not good" as a single word while learning the pattern, the probability of whole sentence can be written as follows;

Probability(P) P(``*phone Service was not good*")

$$= P(``Service" | \text{start of sentence}) * P(``Service|phone")$$
$$* P(``was|Service") * P(`` \text{not}|is") * P(``good|not") \qquad (21)$$

As per maximum likelihood estimation, the condition probability of something like P(``*good|not*") can be given as the ratio of count of the observed occurrence of "not good"

**Table 3** Models Training performance, tested the pre-trained model for the following parameters: setting n = 3-gram, epoch = 10, lr = 0.01 and loss = softmax for fastText model. For the LSVM model, kernel=linear, n = 3 gram.

| Models | Training duration |
|---|---|
| LSVM | 1.2040 Minutes to train 545041 text documents |
| fastText | 0.7699 Seconds to train 545041 text documents |
| SA-BLSTM | 0.9093 Seconds to train 545041 text documents |

together by the count of the observed occurrence of "not". These models can predict new sentences. Similarly, trigram $n = 3$ to learn the probability of occurrence of pattern so that model becomes more accurate. The trigram $n = 3$ results showed more accurate compare to other unigram and bigram testing parameters for both the models LSVM and fastText. The following metrics are used to evaluate the models training performance matrix based on True Positive (TP), True Negative (TN), False Positive (FP) and False Negative (TN).

$$Precision = \frac{tp}{tp + fp} \tag{22}$$

$$Recall = \frac{tp}{tp + fn} \tag{23}$$

$$F1Score = 2 * \frac{Precision * Recall}{Precision + Recall} \tag{24}$$

$$Accuracy\ \% \ = \frac{tp + tn}{tp + fp + tn + fn} * 100 \tag{25}$$

Table 3 shows the model training performance for 3-gram method for both fastText and LSVM. This result shows both linear models were performed well during the model training, fastText performed slightly better than LSVM.

Table 4 shows n-grams performance measures of model's accuracy, Recall, Precision and F1 Score.

The LSVM and fastText are showing similar model accuracy results. SAB-LSTM shows less accuracy. Table 5 shows the % conversations express the positive and negative score of customer sentiment.

## CONCLUSIONS

The results from training the model revealed that fastText performed exceedingly well compared to the LSVM and SA-BLSTM models, and that fastText is much more suitable with large datasets within a server that has minimal configuration. The results revealed that fastText provided a more accurate response within a small duration of time compared with the other two models, obtaining a 90.71% rate in comparison to LSVM and SA-BLSTM

**Table 4  Models test and performance measures results with various parameters.**

| Model | Parameters | Accuracy | Recall | Precision | F1 |
|---|---|---|---|---|---|
| LSVM | Unigram, | 87.74% | 0.88 | 0.88 | 0.88 |
| | Bigram | 89.96% | 0.90 | 0.90 | 0.895 |
| | Trigram | 90.11% | 0.90 | 0.90 | 0.896 |
| | Kernel=linear | | | | |
| fastText | Unigram, | 88.23% | 0.876 | 0.886 | 0.868 |
| | Bigram | 90.55% | 0.896 | 0.907 | 0.901 |
| | Trigram | 90.71% | 0.896 | 0.910 | 0.902 |
| | epoch $= 10$, $r = 0.01$, | | | | |
| | loss $=$ softmax | | | | |
| SA-BLSTM | epoch $= 10$, lr $= 0.01$, | 77.00% | 0.74 | 0.79 | 0.76 |
| | loss $=$ softmax | | | | |

**Table 5  Models sentiment score.**

| Models | Positive sentiment | Negative sentiment |
|---|---|---|
| LSVM | 48.31% | 51.67% |
| fastText | 48.49% | 51.49% |
| SA-BLSTM | 44.67% | 55.31% |

models. The authors concluded that the n-gram method had better compatibility with both fastText and LSVM for the type of dataset used for the experiment, and noticed that training domain-specific datasets improves the accuracy of the sentiment score when tested with a particular domain. fastText shows much better performance during model training compared to the LSVM and SA-BLSTM models. The fastText works well with large dataset within a minimal configuration of server infrastructure setting. The results of the experiment show fastText model training time duration is less, that it gives more accuracy, and that response time is faster than in the LSVM and SA-BLSTM models. The n-gram method works better for both fastText and LSVM, especially trigram $n = 3$ for this dataset. It was noticed that a domain specific dataset training improves the accuracy of the sentiment score when it tested with a particular domain. There is research that is highly essential for the future to explore a framework to build generic models that would be beneficial for industries such as healthcare, retail, and insurance. The SA-BLSTM model the authors have built has the ability to integrate fastText for representation of words to provide increased performance, and has the ability to be pre-trained for such industries that could benefit from this. However, improvements should be made for the quality of audio text files, as well as for the use of automation scripts to correct text errors in conversations.

## Funding

The authors received no funding for this work.

## Competing Interests

The authors declare there are no competing interests.

## Author Contributions

- Anandan Chinnalagu conceived and designed the experiments, performed the experiments, analyzed the data, performed the computation work, prepared figures and/or tables, authored or reviewed drafts of the paper, and approved the final draft.
- Ashok Kumar Durairaj conceived and designed the experiments, analyzed the data, authored or reviewed drafts of the paper, and approved the final draft.

## Data Availability

The Python model files are available in the Supplemental Files.

## Supplemental Information

Supplemental information for this article can be found online at http://dx.doi.org/10.7717/peerj-cs.813#supplemental-information.

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
