# Peer review of "Context-based sentiment analysis on customer reviews using machine learning linear models"

_PeerJ Computer Science, doi:10.7717/peerj-cs.813_

## Round 0.1 · original submission · Minor Revisions

Please revise, as per the reviewers' comments

Reviewer 1 ·

Basic reporting

Clear and unambiguous
Literature review is fair
Figures and Tables are clear

Experimental design

Novelty is lacking only application of existing algorithm with GPU based infrastructure

Validity of the findings

Good choice of algorithm and datasets

Additional comments

1. Paper does not reveal whether multiple modalities have been used for sentiment Analysis. It can be mentioned.
2. Sarcasm is not part analysis and can be added

·

Basic reporting

1. The article is clear in understanding and standard structure has been used.

2. Sufficient literature has been surveyed but lacks standard journal references like IEEE transactions, Elsevier springer etc.

3. The data set used is standard datasets,

Experimental design

1.Sentiment analysis using machine learning is one of the upcoming area and is within aim and scope.

2. Extensive investigation is performed with machine learning and deep learning models.

3. I could see only fasttext comparison with amazon review and all other datasets mentioned, but LSVM and SA-BLSTM of existing literature is not shown for amazon, yelp dataset.

Validity of the findings

1. findings provided in the table are valid.

2. Conclusion should highlight the accuracy measure of the proposed model

Additional comments

Highlight what is the % of performance evaluation in abstract also

---

## Round 0.2 · accepted · Accept

It can be accepted now as the paper is improved

·

Basic reporting

1. The article is clear in understanding and standard structure has been used.

2. Sufficient literature has been surveyed but lacks standard journal references like IEEE transactions, Elsevier springer etc.

3. The data set used is standard datasets,

Experimental design

1.Sentiment analysis using machine learning is one of the upcoming area and is within aim and scope.

2. Extensive investigation is performed with machine learning and deep learning models.

Validity of the findings

1. findings provided in the table are valid.
2. Deep learning method has been tried.

Additional comments

1. All suggestions mentioned in the previous review has been incorporated.